# Local Measures to Curb Dollar Store Growth: A Policy Scan

**DOI:** 10.3390/nu14153092

**Published:** 2022-07-28

**Authors:** Julia McCarthy, Darya Minovi, Chelsea R. Singleton

**Affiliations:** 1Healthy Food, Healthy Lives Program, New York Health Foundation, New York, NY 10018, USA; 2Center for Science and Democracy, Union of Concerned Scientists, Cambridge, MA 02138, USA; dminovi@ucsusa.org; 3Department of Social, Behavioral, and Population Sciences, Tulane School of Public Health and Tropical Medicine, New Orleans, LA 70112, USA; csingle1@tulane.edu

**Keywords:** dollar store, policy scan, moratorium, healthy food, low income, community health

## Abstract

In recent years, advocates have expressed concern about the exponential growth of dollar stores in low-income communities, given their limited stock of healthy foods, and several municipalities in the U.S. have passed novel policies to curb the proliferation of these stores. The purpose of this scan is to create a legal database to inform future healthy retail policies and programs. Legal mapping methods were used to identify local policies aimed at moderating dollar store proliferation. A search yielded 25 policies that met the inclusion criteria, all enacted between 2018 and 2020. Recent policies aiming to slow local dollar store growth were mostly passed in low-income communities of color. All identified policies were passed in either the Midwest or South. The majority of municipalities that passed the policies had populations where more than half of residents identified as non-Hispanic Black or Hispanic and where the poverty rate was greater than the national average. Twelve (48%) municipalities imposed temporary moratoria halting new dollar stores from opening, and ten (40%) banned new construction within a specified distance of an existing dollar store. Key themes identified from analysis of policies’ purpose statements included increasing healthy food availability, diversifying local businesses, and improving community safety. These findings may be useful to leaders in other communities seeking to potentially moderate the impact of dollar stores on community health, as well as researchers and policy makers seeking to evaluate the efficacy of existing policies.

## 1. Introduction

In the United States, dollar stores have grown to be an important source of food for price-sensitive shoppers, especially those with lower incomes [1,2]. Dollar stores, also known as small-box discount stores or discount variety stores, typically sell a wide variety of relatively small and inexpensive items and are limited in size [3]. These stores play an increasingly important role in the retail space, potentially competing with grocery stores. In 2019, sales at the leading dollar store, Dollar General, grew by 20% when the retailer sold more than USD27.7 billion in food, beverages, and other consumable products [4]. Dollar General and its closest competitor Dollar Tree, which also includes Family Dollar, currently operate more than 30,000 outlets across the United States—more than the top ten grocery chains combined [5,6,7,8].

As the number of dollar stores and their share of consumable sales continue to grow, advocates and community leaders have questioned whether dollar stores threaten the economic viability of full-service grocery stores, and ultimately, their community members’ health. As Tulsa City Councilwoman Vanessa Harper-Hall explained, “The community said we don’t want any more dollar stores. We need grocery stores, clothing, shoes—things that you need to live” [9]. Concerned about the growing number of dollar stores in the U.S., policy makers across the country like Harper-Hall have begun introducing policies to temporarily, or permanently, limit the number of new dollar stores that enter their community.

Since 2018, approximately 50 local governments in the U.S. have passed policies to curb dollar store growth. Among these policies are moratoria on new dollar store construction, as well as novel zoning ordinances to limit dollar store proliferation in specific areas of the municipality. By altering permitted, conditional, or prohibited uses in the zoning code, local governments are aiming to (1) limit the number of new dollar stores within a certain radius of existing dollar stores, (2) impose a temporary moratorium on new construction to gather evidence on the public health implications of dollar stores, and/or (3) ban new construction outright.

Many of these policies are framed as public health interventions. The first step to increase the understanding of this phenomenon among public health officials, local policy makers, and researchers is to establish a legal database of dollar store policies. Timely collection and analysis of legal information, information that is essential to the evaluation of policy interventions, is often excluded from the universe of public health research [10]. This policy scan aims to (1) identify local policies that limit the construction of new dollar stores for public health reasons, (2) evaluate the substantive text of these policies, and (3) summarize their legal provisions and stated purposes. The goal of this legal database is to help public health practitioners determine which jurisdictions (if any) have effective laws and, if effective, which elements of the laws work best.

## 2. Materials and Methods

### 2.1. Policy Identification

Policies limiting the growth of dollar stores are a relatively new phenomenon about which the public lacks adequate information. Legal mapping is a strategy that can address this knowledge gap [10]. It requires the systematic collection, analysis, and dissemination of policies of significance implemented within a defined period and geography. Aggregating legal information into a database allows public health officials, policy makers, advocates, and community members to compare and evaluate key features of laws designed to improve public health. In this case, legal mapping can provide timely information on the status of emerging healthy retail policies.

There is no single clearinghouse for municipal or county codes, which makes policy scans of this nature more difficult than those at the state and federal levels. Local policies designed to slow down dollar store growth are so new that many municipal and county codes have yet to be updated with such text. Given the novelty and difficulty of finding such policies, our research team first used Google to search for news articles published after 1 January 2018 announcing dollar store policies. Search terms included “dollar store restrictions,” “dollar store dispersal,” “dollar store moratorium,” “dollar store ordinance,” “limiting dollar stores,” “small-box store restrictions,” “small-box store dispersal,” “small-box store moratorium,” “small-box store ordinance,” “limiting small-box stores,” “discount store restrictions,” “discount store dispersal,” “discount store moratorium,” “discount store ordinance,” and “limiting discount stores.”

We cross-referenced our results with the gray literature (such as reports on existing retail restrictions from local governments or the Institute for Local Self-Reliance, a non-profit organization that tracks policies designed to promote local business) and Municode, a platform that publishes legal documents for local governments [11,12]. A March 2020 search yielded 29 distinct dollar store policies. We obtained copies of each policy from local government websites and officials and conducted an initial assessment of document text. From the 2020 scan, we determined that 18 of the 29 were relevant to this scan. Relevant policies were those that (1) limited dollar store expansion within the municipality, (2) were passed between January 2018 and the date of the search, and (3) had language in the purpose statement or media quotes from the policy sponsors suggesting the policy aimed to improve community and/or public health. In February 2021, we expanded our inclusion criteria to consider policies passed from January 2018 to 31 December 2020. Our updated search yielded 7 additional policies, bringing the final sample to 25.

### 2.2. Document Analysis and Geographic Data

We performed a qualitative document analysis of the 25 municipal and county policies that met our inclusion criteria to characterize the legal mechanism and purpose of the legislation. We reviewed the substantive provisions of each policy to determine the types of retail outlets covered, the legal mechanism employed, the time period of restriction, and any incentives included to encourage healthier food environments. We grouped the policies into four categories that describe the legal mechanism used to prevent dollar store proliferation: moratoria, special exceptions, prohibited uses, and overlays. Definitions and examples of each legal mechanism are provided in Table 1.

Statements of purpose or intent were most often available in the bill’s preamble or other relevant policy documents. Members of the research team examined the text of all legislative documents (bill text, committee reports, meeting notes, etc.) to develop an initial list of codes and emerging themes regarding legislation purpose. Afterward, two separate reviewers (J. E. M. and C. R. S.) independently coded the text and evaluated the language on the legislation’s purpose. Both reviewers used this language to group municipalities based on emerging themes. When reviewers disagreed on the purpose(s) of a specific policy, a third reviewer (D. M.) resolved the discrepancy.

We extracted socio-demographic and environmental data from the U.S. Census Bureau on all 25 municipalities [13]. The measures collected represent the 2019 American Community Survey 5-Year estimates and include the following: total population size, % non-Hispanic Black residents, % Hispanic residents, % residents living below the federal poverty level, and size of municipality (in square miles). We also obtained data on the total number of dollar stores in each municipality from Reference Solutions [14]. We used these data to calculate the total number of dollar stores per square mile for each municipality.

## 3. Results

### 3.1. Municipality Characteristics

Socio-demographic and environmental characteristics of the municipalities included in the scan are provided in Table 2. Of the 25 dollar store policies identified, 14 (56%) were from municipalities in Southern states and 11 (44%) in Midwestern states. Two (8%) were adopted at the county level and twenty-three (92%) at the town/city level. Twelve municipalities (48%) had population sizes greater than 100,000 and were labeled “urban” according to the U.S. Census. The percentage of non-Hispanic Black residents ranged from 0.8% (Palm City, FL) to 93.6% (Stonecrest, GA), while the percentage of Hispanic residents ranged from 1.6% (Broadview Heights, OH) to 44.6% (Baytown, TX). Twenty-two municipalities (88%) had a percentage of non-Hispanic Black residents higher than the national average of 13.4%. The percentage of residents experiencing poverty ranged from 1.9% (Broadview Height, OH) to 34.6% (Cleveland, OH). Nineteen municipalities (76%) had a percentage of impoverished residents higher than the national average of 10.5%. The number of dollar stores in the municipalities that passed the policies ranged from zero (Palm City, FL) to 88 (Oklahoma City, OK). The number of dollar stores per square mile ranged from zero (Palm City, FL) to 1.03 (Cleveland, OH).

### 3.2. Relevant Policy Provisions

Table 3 provides a summary of the relevant policy provisions for all 25 municipalities that passed dollar store policies. Twelve (48%) passed moratoria banning permits for new stores. These moratoria ranged in length from 120 days to 19 months. Two of the twelve moratoria included conditions that waived the restriction for new dollar stores that planned to dedicate at least 15% of their shelf space to fresh foods (e.g., fresh fruits, vegetables, meat, dairy). Three municipalities (12%) implemented permanent policies that prohibited the development of new dollar stores in either the entire municipality or a demarcated overlay district. These overlay districts were identified as areas of need with limited access to retailers that offer healthy foods. The remaining 10 municipalities (40%) passed permanent policies that prohibited new dollar store development within a specified distance of an existing dollar store. This distance ranged from 2500 feet to 2 miles. Four of the ten included conditions that waived the ban for new stores that planned to dedicate at least 15% of the store’s shelf space to fresh foods. Two of the ten prohibited development only in demarcated overlay districts. Three municipalities included provisions to support food retail outlets other than dollar stores, for example, reducing the parking requirement typically required of grocery stores or allowing produce sales from community gardens.

### 3.3. Purpose of Legislation

Findings from the document analysis identifying the purpose(s) of each dollar store policy are provided in Table 4. Six themes emerged from the document analysis: (1) address the lack of healthy food options offered by dollar stores, (2) expand the diversity of retail food stores in the municipality, (3) support the local economy and businesses in the municipality, (4) improve community safety and prevent further blight, (5) enhance community aesthetics (i.e., beauty), and (6) address dollar store labor and cost concerns.

Approximately 80% of the policies sought to increase the number of fresh food retailers in the area. In total, 19 of the 25 policies (76%) included language that suggested the goal was to address the lack of healthy food offerings in dollar stores. Of these, all but one discussed how preventing dollar store proliferation could help diversify food retailers in the municipality.

Ten policies (40%) indicated that improving community safety and/or preventing further blight was a goal, and nine (36%) aimed to support the local economy and businesses (e.g., local grocery stores). Policies that described community safety as a purpose mentioned topics such as preventing crime and theft in and near dollar stores. Seven policies (28%) included language on preserving community aesthetics, and four policies (16%) included language on addressing dollar store labor and cost concerns. Examples of labor and cost concerns mentioned include the employment of fewer people at a lower wage than local grocery stores and the high price of individual-sized packaged foods after accounting for price per ounce.

## 4. Discussion

The aim of this policy scan was to identify and examine legislation that curbs dollar store proliferation for public health reasons. Local governments routinely use policies to promote health, but these policies are rarely tracked systematically and are even less frequently evaluated [10]. A database of existing dollar store policies enables researchers to evaluate the efficacy of existing policies. It provides public health advocates looking to partner with community groups with examples of community-led interventions. Most importantly, it empowers leaders interested in improving the healthfulness of retail options in their own communities, offering potential legislative models.

Concentrated in the South and Midwest, many of the policies limiting dollar stores’ spread were introduced by policy makers who reflect the municipality’s demographics, underscoring the growing concern with inadequate, unhealthy food retail options among historically disenfranchised communities. As one councilman pushing for a policy to limit dollar store growth explained, “Just because we’re poor and communities of color doesn’t mean that we can’t demand quality” [15].

Low-income communities of color, where dollar stores are often densely concentrated, are significantly less likely to have full-service grocery stores that offer high-quality affordable produce compared with higher-income and majority-White communities [16,17]. These communities are also more likely to have a high prevalence of small food retailers (e.g., corner stores, dollar stores, liquor stores) [18]. Smaller stores typically carry a limited supply of healthy foods, many of which are of lower quality, as well as a wide range of high-calorie convenience items (e.g., snacks, candy, and sugar-sweetened beverages) [16,17]. The disparities in the quantity and quality of food retailers in low-income communities of color are often attributed to a history of discriminatory actions, including redlining, exclusionary zoning, and other forms of chronic underinvestment [19,20]. Thus, the issue of food retailer quality and diversity is closely tied to the racial/ethnic and socio-economic inequities in health and nutrition in the U.S.

Critics of dollar stores assert that these retailers have capitalized on the retail void in low-income communities of color, offering a restricted range of products where competition was already sparse [19,21]. Unlike full-service grocery stores, the food and beverages sold at dollar stores are largely ultra-processed and limited in selection [22]. A prior study of consumer purchasing patterns at small food retailers found that individuals shopping at dollar stores were the most likely to purchase SSBs and candy [1]. On the flip side, research shows that individuals residing in communities with better access to supermarkets and limited access to stores that offer few healthy food options have healthier diets [23]. When more fresh produce is available, consumers purchase more fruits and vegetables and fewer sugar-sweetened beverages [24,25]. These studies support the hypothesis that customer purchases align with food retailers’ stock and suggest the entrance of dollar stores may not support a healthy diet.

The number and popularity of dollar store policies suggest that a localized model of civic engagement, which involves community organizers and diverse stakeholders, could be useful in current and future efforts to improve the food environment. For example, our research shows that there are robust examples of policies created and successfully passed in communities with significant Black and low-income populations. In total, 21 of the 25 jurisdictions with dollar store policies have Black populations higher than the national average, and 19 of the 25 jurisdictions exceed the national average for the percentage of the population living in poverty.

Understanding the factors supporting policy development could be key to engaging communities and centering their voices in future public health policy development. Three major themes regarding policy purpose that emerged were (1) address the lack of fresh foods offered by dollar stores, (2) increase the diversity of local food retailers in the municipality, and (3) support local businesses that offer fresh foods (i.e., grocery stores). These results suggest that expanding healthy food retail is a key motivating force behind these policies. Several adverse outcomes, including childhood obesity, have been linked to the quality of food environments [26]. Theoretical frameworks, including the Planning Healthy Cities Conceptual Framework developed by Northridge and colleagues, highlight the significance of improving healthy retail options in local food environments to further health promotion endeavors [27,28]. Many of the policies included provisions that barred new construction in overlay areas of concern (e.g., low-income/low food access communities) or permitted construction if the store allocated a certain percentage of floor space to fresh foods.

Other emerging themes include improving community safety/blight, enhancing community aesthetics, and addressing labor/cost concerns. Most policies citing safety or blight as a purpose explicitly mentioned the issue of crime, another social determinant of health, in and near dollar stores in their municipality. Several studies have reported that food retailers, especially small retailers (e.g., liquor stores, corner stores, dollar stores), are crime attractors [29,30]. Small retailers often lack adequate security measures and have few employees, which increases the opportunity for crime occurrence [29]. Considering that crime is disproportionately higher in low-income communities of color, safety concerns may prove to be a significant factor in the development of additional dollar store policies in the U.S. [31].

While concern for community character was not a predominant theme in the policies included in this scan, a desire to protect and enhance community aesthetics is a common underpinning of traditional formula business restrictions—policies that restrict the growth of chain stores [32]. Many of the policies that cited community aesthetics expressed a concern that small box discount stores were changing the character of these municipalities. While not directly related to social determinants of health, the preservation of community aesthetics may contribute to lower perceived stress among residents [33]. Examples of labor and cost concerns mentioned include the employment of fewer people at a lower wage than local grocery stores and the high price of individual-sized packaged foods after accounting for price per ounce. There is a substantial body of evidence linking both working conditions and food prices with human health [34,35]. Specifically, lower wages and higher relative food costs are associated with limited access to basic needs, including healthy food [36]. Using their permit and licensing authority to promote better labor practices and raise awareness about relative food costs are two ways that local governments can influence public health and potentially reduce health inequities.

The research has strengths and limitations. To our knowledge, this is the first study to evaluate the provisions and purpose of legal measures to slow down dollar store proliferation in the U.S. By employing legal mapping methods, we were able to identify and obtain legal documents for recently enacted policies. Unfortunately, the lack of a central database on local laws and policies challenged our ability to conduct a thorough search. We relied heavily on gray literature, including articles published by local media outlets, to identify these policies. It is possible that we did not include some municipalities in our scan. Future studies should consider this limitation if taking a similar approach to reviewing local policies.

Furthermore, additional research is necessary to evaluate the extent to which these policies achieve their stated purpose, particularly increasing access to healthier food options. This policy scan provides a unified location to access policy information and analysis and can serve as a foundation for future policy surveillance, the systematic, scientific collection and analysis of laws of public health significance [10]. However, this scan does not address which jurisdictions have effective dollar store laws and, if effective, which elements of the laws work best. Research is needed to evaluate the extent to which these dollar store policies increase healthy food access, as well as the other stated purposes of these policies. For example, future research could examine whether such policies impact grocery store entrance and success or whether having fewer retail options is better than having multiple dollar stores. Additional research could also include a survey of key organizations that track local nutrition and food policies to determine the landscape into which these policies fit. Most importantly, more research is needed to understand community motivation to limit dollar store growth, particularly in low-income communities of color. Public health officials and advocates could use such information to support future community-led public health interventions in communities that have historically had fewer public health protections.

## 5. Conclusions

Public health professionals have long studied policies aiming to expand healthy food retail in low-income communities, but there is little research on policies that reduce access to retailers that offer mostly unhealthy food items. Policies restricting dollar store development, which are currently subject to much public discussion and policy-maker consideration, are growing in popularity among low-income communities and communities of color. The number and recent popularity of these policies suggest that there are localized and politically powerful models of civic engagement that warrant further study. Analyzing the prevalence of and variations among policies that restrict dollar store proliferation can ultimately deepen public health officials’, advocacy organizations’, and funders’ ability to support community participation, empowerment, and action. Furthermore, given the field’s and nation’s growing interest in dismantling structural barriers in historically disenfranchised communities to maintaining a healthy diet, these policies may be important for future upstream initiatives to address inequities in consumer food purchasing, dietary intake, and community engagement.

## Figures and Tables

**Table 1 nutrients-14-03092-t001:** Definitions of legal mechanisms and examples of key legal terms.

Mechanism	Definition	Example Text
**Moratorium**	A ban on rezoning or permit applications. Moratoria are frequently imposed to preserve the status quo before adopting a new zoning ordinance.	*Cleveland Ord. No. 411-2019*. This Council establishes a moratorium on the review and issuance of zoning permits, certificates of occupancy, and other license or permit applications for small box discount stores until 31 December 2020 or until such time as the City has established regulations regarding review and issuance of small box discount stores, whichever is earlier
**Special Exception**	A land use permitted with local government approval. The approval process involves ensuring the standards outlined in the ordinance are met, usually after a public hearing. The terms “special use” and “conditional use” are often used interchangeably with “special exception,” the only difference being that the jurisdiction’s governing body rather than a zoning board must approve “conditional uses.”	*Fort Worth Ord. No. 21,653.* A small box discount store may be permitted in accordance with the use tables in Chapter 4, Articles 6, 8 and 12. The City Council may consider the following criteria: (a) the proposed location is no less than 2 miles from any existing small box discount store (b) a minimum of 10% of the floor area is dedicated to fresh produce, meat, and dairy products
**Prohibited Use**	Any use of land, which is not specifically listed as a use or special exception within a zoning district. Local officials may grant a variance, allowing owners to use the land in a manner normally prohibited if the owner demonstrates undue hardship.	*Baytown Ord. No. 14,380.* Any of the following land uses shall not occur unless it is approved by city council as part of a planned unit development (PUD): Small box discount retail
**Overlay District**	A district applied over one or more existing zoning districts, typically to establish additional or stricter standards for new properties.	*Birmingham Ord. No. 15-133*. The intent of this Article is to establish a Healthy Food Overlay District for the City of Birmingham. The regulations of this section apply to all new uses and structures within the boundaries of the Healthy Food Overlay District…mapped using the low-income/low access census tract data identified as food deserts by the USDA

**Table 2 nutrients-14-03092-t002:** Socio-demographic and environmental characteristics of municipalities (N = 25).

Municipality	Policy Year	Population Size ^a^	% Black ^a^	% Hispanic ^a^	% Poverty ^a^	Land Area (sq. mile) ^b^	Number of Dollar Stores	Dollar Stores (per sq. mile)
Birmingham, AL	2019	209,403	71	4	27	146.1	55	0.38
Palm City, FL	2020	23,120	1	6	6	13.9	0	0
Atlanta, GA	2019	506,811	52	4	22	133.2	81	0.61
College Park, GA	2020	15,159	80	4	30	10.1	2	0.20
Clarkston, GA	2020	12,637	60	4	31	1.1	4	3.64
DeKalb County, GA	2020	759,297	55	9	14	267.6	68	0.25
East Point, GA	2020	34,875	76	9	22	14.7	6	0.41
Stonecrest, GA	2019	54,522	94	2	19	37.2	5	0.13
Wyandotte County, KS	2019	491,918	29	10	17	315.0	1	0.003
New Orleans, LA	2019	391,006	60	6	25	169.4	46	0.27
Melvindale, MI	2019	10,248	14	22	28	2.7	2	0.74
Southfield, MI	2020	72,689	69	2	11	26.3	12	0.46
Akron, OH	2019	197,597	30	2	23	62.0	46	0.74
Broadview Heights, OH	2019	19,102	3	2	2	13.1	1	0.08
Brunswick, OH	2020	34,880	3	3	8	12.9	3	0.23
Cleveland, OH	2019	381,009	50	12	35	77.7	80	1.03
North Royalton, OH	2020	30,068	2	2	5	21.3	1	0.05
Toledo, OH	2020	272,779	27	9	26	80.7	48	0.59
Oklahoma City, OK	2019	649,021	15	19	17	606.4	88	0.15
Tulsa, OK	2018	400,669	15	16	20	196.8	56	0.28
Mauldin, SC	2020	25,409	25	9	8	10.0	3	0.30
Baytown, TX	2020	77,192	19	45	16	35.5	17	0.48
Fort Worth, TX	2019	895,008	19	35	16	399.8	76	0.19
Manvel, TX	2020	12,671	20	34	4	23.5	1	0.04
Mesquite, TX	2018	142,816	26	40	13	46.0	19	0.41

sq. = square. Note: Municipality represents a city or county with a governing body and local government. ^a^ Population size, % Non-Hispanic Black, % Hispanic, and % poverty represent 2019 American Community Survey estimates from the U.S. Census Bureau: www.census.gov/quickfacts/fact (accessed on 23 September 2020). ^b^ Land area in square miles represents 2010 estimates from the U.S. Census Bureau: www.census.gov/quickfacts/fact (accessed on 23 September 2020).

**Table 3 nutrients-14-03092-t003:** Summary of relevant policy provisions (N = 25).

Municipality	Defining Size of Dollar Store (sq. feet)	Legal Mechanism(s)	Time Period	Limitation(s)	Healthy Food Incentives
Palm City, FL	<16,000	moratorium	120 days	-	-
Clarkston, GA	7500–16,000	moratorium	until 9/21/20 (~6 months)	-	-
DeKalb County, GA	<16,000	moratorium	45 days; extended an additional 180 days	-	-
East Point, GA	<16,000	moratorium	120 days	-	-
Southfield, MI	5000–15,000	moratorium	180 days	-	-
Broadview Heights, OH	<15,000	moratorium	12 months	-	-
Brunswick, OH	3000–15,000	moratorium	6 months; extended an additional 6 months	-	-
Cleveland, OH	3000–15,000	moratorium	19 months	-	Waives conditions for stores that dedicate at least 15% of shelf space to fresh foods and produce
North Royalton, OH	3000–15,000	moratorium	12 months	-	-
Toledo, OH	<15,000	moratorium	until 12/31/20 (~7 months)	-	-
Mauldin, SC	<10,000	moratorium	180 days	-	Waives conditions for stores that dedicate at least 15% floor area to fresh foods and vegetables
Oklahoma City, OK	<12,000	moratorium with special exception	180 days	within 1 mile of an existing dollar store	-
College Park, GA	<15,000	prohibited use	permanent	-	-
Stonecrest, GA	<12,000	prohibited use (overlay)	permanent	-	-
Baytown, TX	<12,000	prohibited use (overlay)	permanent	-	-
Birmingham, AL	<12,000	special exception (overlay)	permanent	within 1 mile of an existing dollar store	Reduces parking requirement for grocery stores; allows produce sales in community gardens
Atlanta, GA	<12,000	special exception	permanent	within 1 mile of an existing dollar store	-
Wyandotte County, KS	<15,000	special exception	permanent	within 10,000 ft. of an existing dollar store or 200 ft. of a residence	Waives conditions for stores that dedicate at least 15% of shelf space to fresh or fresh frozen food
New Orleans, LA	5000–15,000	special exception	permanent	within 2 miles of an existing dollar store in New Orleans East and the West Bank and within 1 mile of an existing dollar store elsewhere	Waives conditions for stores that dedicate at least 15% of shelf space to fresh or fresh frozen food; entitles grocery stores that dedicate 30% or more of shelf/display area to fresh/fresh frozen foods to an additional 5000 sq. ft
Melvindale, MI	<12,000	special exception	permanent	within 2500 ft. of an existing dollar store	Waives conditions for stores that dedicate at least 15% of floor area to fresh produce, meat, and dairy
Akron, OH	NS	special exception	permanent	within 2500 ft. of an existing dollar store	-
Tulsa, OK	<12,000	special exception (overlay)	permanent	within 1 mile of an existing dollar store	Reduces parking requirement for grocery stores; allows produce sales in community gardens
Fort Worth, TX	<10,000	special exception	permanent	within 2 miles of an existing dollar store	Waives conditions for stores that dedicate at least 15% of floor area to fresh foods and vegetables
Manvel, TX	NS	special exception	permanent	within 2 miles of an existing dollar store and a minimum of 10% of floor space dedicated to fresh produce, meat, and dairy products	-
Mesquite, TX	N/A	special exception	permanent	within 5000 ft. of an existing dollar store	-

-: not included; NS: not specified in policy documentation; sq: square; ft: feet. Note: Municipality represents a city or county with a governing body and local government.

**Table 4 nutrients-14-03092-t004:** Stated purpose of legislation by municipality (N = 25).

Municipality	Overarching Objective of Legislation	Purpose of Legislation					
		Address Lack of Healthy Food Options	Expand Diversity of Retail Stores	Improve Community Safety/Blight	Support Local Economy and Businesses	Enhance Community Aesthetics	Address Labor and Cost Concerns
Birmingham, AL	Modify existing regulations to allow for more diverse retail options and convenient access to fresh meats, fruits, and vegetables	X	X				
Palm City, FL	Mitigate negative secondary effects of use on public health, safety, and welfare	X	X	X	X	X	
Atlanta, GA	Prevent proliferation in economically depressed areas with scarce access to healthy and affordable food options	X	X	X			
College Park, GA	Prevent economically depressive state of neighborhoods and diminishment viability of supermarkets		X	X			
Clarkston, GA	Protect the public health, welfare, and aesthetics of the city	X	X			X	
DeKalb County, GA ^a^	Study the effects of small box discount stores on health, safety, and welfare						
East Point, GA	Study the effects on health, safety, and welfare of the city’s residents and businesses				X	X	
Stonecrest, GA	Address the economically depressive state of neighborhoods and diminishment viability of alternative options	X	X	X	X		
Wyandotte County, KS	Regulate availability to assure the best possible opportunity to provide fresh fruits and vegetables to the community	X	X	X	X	X	
New Orleans, LA	Address rapid proliferation that may impede the successful entry of full-line grocery stores	X	X			X	
Melvindale, MI	Regulate the proliferation of stores in the city and improve opportunities to offer fresh healthy foods	X	X				
Southfield, MI	Prevent potential negative effects on the business of the city’s existing supermarkets	X	X				
Akron, OH	Guard against local business disinvestment within neighborhoods that lack access to fresh food	X	X	X	X		X
Broadview Heights, OH	Address proliferation to protect the health, safety, and welfare of community members	X	X	X	X		X
Brunswick, OH ^a^	Preserve the public health, safety, and general welfare of the city’s residents and property owners						
Cleveland, OH	Preserve public peace, property, health, safety, and welfare	X	X	X	X		X
North Royalton, OH ^a^	Preserve the public peace, health, safety, and welfare of the city and review the city’s proposed master plan						
Toledo, OH	Study the impact on public health and safety	X		X			
Oklahoma City, OK	Regulation to preserve property values, prevent blight, and protect the health, safety, and general welfare of the residents	X	X	X			
Tulsa, OK	Reduce over-concentration of small box discount stores to increase diversity of retail activity and allow for community-based approaches to distributing healthy foods	X	X		X		
Mauldin, SC	Study the effects on local business, job growth, and access to fresh foods	X	X		X	X	X
Baytown, TX	Improve the city’s image and recruit higher-end retail establishments that provide retail diversity	X	X			X	
Fort Worth, TX	Promote access to healthy food options in underserved neighborhoods	X					
Manvel, TX	Expand the diversity of retail businesses in the city	X	X				
Mesquite, TX	Promote the availability of fresh and quality foods, especially in underserved neighborhoods	X	X				
**TOTAL, n (%)**		**20 (80%)**	**19 (76%)**	**10 (40%)**	**9 (36%)**	**7 (28%)**	**4 (16%)**

X: stated policy purpose. ^a^ Purpose statement for this policy only includes sponsor language of “improving public health, safety, and welfare”. No themes were identified.

## Data Availability

Not applicable.

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
