# Peer review of "Local Measures to Curb Dollar Store Growth: A Policy Scan"

_nutrients, 2022, doi:10.3390/nu14153092_

Round 1
Reviewer 1 Report
The study itself is well done, the data collection seems sound, and the results that are reported, are reported well. However, the results are limited so the introduction and discussion need to address the following.
The entire study begs the question of how any of these policies would accomplish their goals. Just because a community bans Dollar Stores does not automatically result in increasing healthy food, and how would it reduce blight? Isn’t it better to have a functioning store than an empty lot? And aesthetics? Are dollar stores uglier than other stores? Did the policies provide funding for other types of retailers or zoning regulations to attract other types of retailers? The problem is the results are just not useful practically without more information. If the authors are unable to add this, their study should be explained as a very limited review of only the policies to get rid off dollar stores without any information on how these policies will address community needs or accomplish any actual goals. It is unclear what information a policymaker can use from this. If this reviewer is missing how this adds to the literature but the authors can identify an answer to that, this should be explicitly explained. Then the issues I have identified should be in the discussion as need for future research and limitations of this study.
The introduction/discussion: There does not seem to be evidence dollar stores are inherently problematic or harmful to communities. The evidence presented anti-dollar stores in the introduction is one-sided and seems to discount the position that these locations had no stores so the dollar store provided something missing in the community. Sales are increasing! There is no indication that Full service grocery stores would have moved in instead or were thwarted from doing so. Lines 61-64 about reference 1 do not add anything to the paper. If they don’t sell fresh fruit and vegetables then of course people are not purchasing those at dollar stores. Do they sell fresh fruit and vegetables? The authors should rewrite the introduction to be less subjective about dollar stores and consider presenting information about why legislators say they are proposing the bills and what legislators proposing the bills seek to accomplish. This would be more in line with this study which is not about the offerings at Dollar Stores but rather the laws. Then the discussion would be a place to present additional information. Need to define small box discount stores.Author Response
Please see attachment.

Reviewer 2 Report
I am in the rare position of having very few comments for the authors. This scan of policies that restrict dollar stores is timely, novel, well-written, clear, and sound in methodology. I have only a few minor comments below:
Results
-Aside from store size, was other criteria were used to define "dollar stores" in legislation?
-Do the authors have information they could share in the manuscript on which policies were enacted via city council or other legislative body vs. through a voter or ballot initiative? Not necessary but helpful if this info is handy.
-Table 2: Are the 2019 ACS estimates 5-yr, 3-yr, or other estimates?
-Table 4: Missing a count in the far right column at the bottom. Percentage is listed though.
Discussion
-A recommendation for future policy scans could be to survey key organizations that track local nutrition and food policies.
Supplemental material
-It would be helpful for the reader to have a table that lists the ordinance numbers and provides links to any PDFs of the ordinances. This is especially helpful for policymakers and advocates who may want to borrow language from prior policies.
